# Learning to recover orientations from projections in single-particle cryo-EM

## Abstract

A major challenge in single-particle cryo-electron microscopy (cryo-EM) is that the orientations adopted by the 3D particles prior to imaging are unknown; yet, this knowledge is essential for high-resolution reconstruction. We present a method to recover these orientations directly from the acquired set of 2D projections. Our approach consists of two steps: (i) the estimation of distances between pairs of projections, and (ii) the recovery of the orientation of each projection from these distances. In step (i), pairwise distances are estimated by a Siamese neural network trained on synthetic cryo-EM projections from resolved bio-structures. In step (ii), orientations are recovered by minimizing the difference between the distances estimated from the projections and the distances induced by the recovered orientations. We evaluated the method on synthetic cryo-EM datasets. Current results demonstrate that orientations can be accurately recovered from projections that are shifted and corrupted with a high level of noise. The accuracy of the recovery depends on the accuracy of the distance estimator. While not yet deployed in a real experimental setup, the proposed method offers a novel learning-based take on orientation recovery in SPA. Our code is available at `https://github.com/anonymous/protein-reconstruction`.

## 1 Introduction

Single-particle cryo-electron microscopy (cryo-EM) has revolutionized the field of structural biology over the last decades [1, 2, 3]. The use of electron beams to image ice-embedded samples has permitted the recovery of 3D bio-structures at unprecedented resolution. This "resolution revolution" has had a tremendous impact in biomedical research, providing invaluable insights into the biological processes that underlie many current diseases.

In single-particle cryo-EM, every 3D particle adopts a random orientation $\boldsymbol{\theta}_i$ in the ice layer before being imaged. Hence, the projection geometry associated to each acquired 2D projection (Figure 1) is unknown. Yet, this knowledge is essential for the tomographic reconstruction of bio-structures [4]. We consider that a cryo-EM measurement (*i.e.*, a projection) $\mathbf{p}_i \in \mathbb{R}^{n_p}$ is acquired through

$$\mathbf{p}_i = \mathbf{C}_{\boldsymbol{\varphi}} \mathbf{S}_{\mathbf{t}_i} \mathbf{P}_{\boldsymbol{\theta}_i} \mathbf{x} + \mathbf{n}, \tag{1}$$

where $\mathbf{x} \in \mathbb{R}^{n_x}$ is the unknown 3D density map [5] (Coulomb potential). The operator $\mathbf{P}_{\boldsymbol{\theta}_i} : \mathbb{R}^{n_x} \to \mathbb{R}^{n_p}$ is the projection along the orientation $\boldsymbol{\theta}_i$ (*i.e.*, the x-ray transform). The operator $\mathbf{S}_{\mathbf{t}_i} : \mathbb{R}^{n_p} \to \mathbb{R}^{n_p}$ is a shift of the projection by $\mathbf{t}_i = (t_{i_1}, t_{i_2})$. The convolution operator $\mathbf{C}_{\boldsymbol{\varphi}} : \mathbb{R}^{n_p} \to \mathbb{R}^{n_p}$ models the microscope point-spread function (PSF) with parameters $\boldsymbol{\varphi} = (d_1, d_2, \alpha_{\text{ast}})$, where $d_1$ is the defocus-major, $d_2$ is the defocus-minor, and $\alpha_{\text{ast}}$ is the angle of astigmatism [6, 7]. Finally, $\mathbf{n} \in \mathbb{R}^{n_p}$ represents additive noise. Figure 11 illustrates the effect of projection, shift, and noise. The challenge is then to reconstruct $\mathbf{x}$ from a set of projections $\{\mathbf{p}_i\}_{i=1}^{P}$ acquired along unknown orientations.

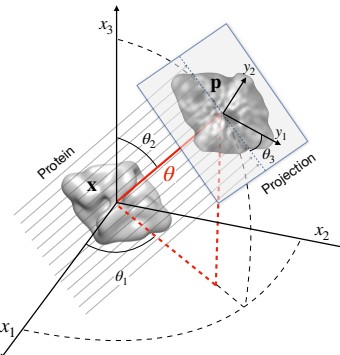

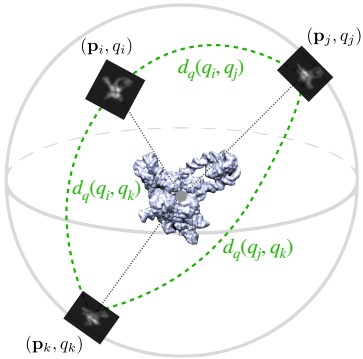

Figure 1: Geometry of the imaging model defined in (1). The 3D density $\mathbf{x}$ in the coordinate system $(x_1, x_2, x_3)$ is imaged along the *orientation* $\boldsymbol{\theta}$ to produce the 2D *projection* $\mathbf{p}$ in the coordinate system $(y_1, y_2)$ of the microscope's detector plane. The orientation $\boldsymbol{\theta} = (\theta_3, \theta_2, \theta_1)$ is decomposed as the direction $(\theta_2, \theta_1) \in [0, \pi] \times [0, 2\pi[$ (parameterizing the sphere $\mathbb{S}^2$) and the in-plane rotation $\theta_3 \in [0, 2\pi[$ (parameterizing the circle $\mathbb{S}^1$). In our work, we represent the orientation $\boldsymbol{\theta}$ as a unit quaternion $q$.

Figure 2: Single-particle cryo-EM produces $P$ projections (with $P$ in the order of $10^5$) from unknown orientations: $\{(\mathbf{p}_i, q_i)\}_{i=1}^P$. Observing that distances between orientations constrain the latter, we aim to *recover the orientations* $\{q_i\}$ from $\{d_q(q_i, q_j)\}$, where $d_q(q_i, q_j)$ is the distance (angle) between orientations $q_i$ and $q_j$. Observing that the similarity between projections depends on their relative orientation, we aim to *estimate the distance* $d_q(q_i, q_j)$ from the projections $(\mathbf{p}_i, \mathbf{p}_j)$.

A popular approach is to alternatively refine the 3D structure and estimated orientations [8, 9, 10, 11, 12, 13]. Yet, the outcome of these iterative-refinement procedures is often predicated on the quality of the initial reconstruction, or, equivalently, on the initial estimation of the orientations [14, 15].

Several methods have been designed to produce a first rough *ab initio* structure for the refinement procedure [16]. *Moment-matching* techniques [17, 18, 19, 20] reconstruct an initial structure such that the first few moments of the distribution of its theoretical measurements match the ones of its experimental projections; however, they typically remain sensitive to error in data and can require relatively high computational complexity. Based on the central-slice theorem, *common-lines* methods [21, 8, 22, 23, 24, 25, 26] aim at uniquely determining the orientations of each projection by identifying the common-lines between triplets of projections—a real challenge given the massive amount of noise. Alternatively, the marginalized maximum likelihood (ML) formulation of the reconstruction problem [11]—classically used for the iterative-refinement procedures themselves—can be minimized using stochastic gradient descent [27]. This permits to avoid the need for an initial volume estimate, at the possible cost of greater convergence instability.

More recently, the recovery of geometrical information from unknown view tomography of 2D point sources has been proposed [28], but the extension to 3D cryo-EM tomography is not straightforward. Finally, [29] proposed to recover the in-plane rotations by learning to embed projections in an appropriate latent space, but only after directions had been estimated through three rounds of 2D classification in RELION.

Despite the aforementioned advances, providing a robust initial volume remains a challenge due to the high-dimensionality and ill-posedness of the underlying optimization problem. On the other hand, the remarkable ability of convolutional neural networks to capture relevant representations of images has had a profound influence in imaging [30]. In this work, we present a learning-based approach to recover the unknown orientations directly from the acquired set of projections—without the need for an intermediate reconstruction procedure or an initial volume estimate.

## 2 Method

Our approach relies on two observations (Figure 2), yielding two steps (Figure 3). First, the more similar two projections $(\mathbf{p}_i, \mathbf{p}_j)$, the more likely they originated from two particles that adopted close

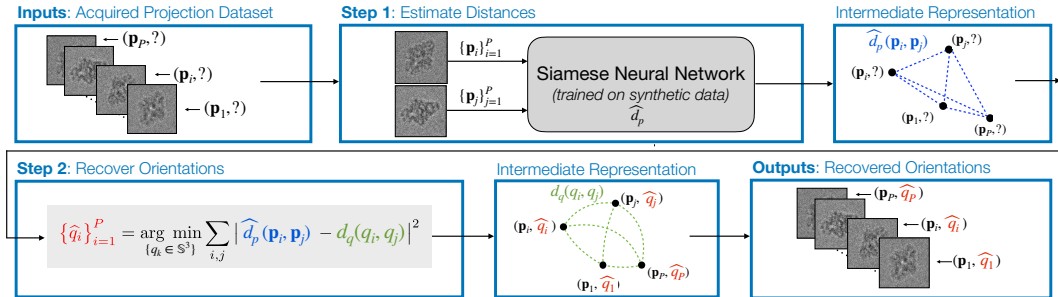

Figure 3: Our method consists of two steps. First, we estimate distances between pairs of projections. Second, we recover the orientation of each projection from these distances.

orientations $(q_i, q_j)$ in the ice prior to imaging;[1] this observation guides a number of applications in the field [2]. Hence, we aim to *estimate distances* between orientations $d_q(q_i, q_j)$ from the projections as $\widehat{d_p}(\mathbf{p}_i, \mathbf{p}_j)$, which we discuss in §2.2. Second, an orientation $q$ is constrained by the distances between itself and the other orientations $\{d(q, q_j)\}$. Hence, we aim to *recover orientations* $\{\widehat{q_k}\}$ such that the induced distances $\{d_q(\widehat{q_i}, \widehat{q_j})\}$ are close to the estimated distances $\{\widehat{d_p}(\mathbf{p}_i, \mathbf{p}_j)\}$, which we discuss in §2.3. All in all, from a set of projections $\{\mathbf{p}_k\}$, we aim to recover their orientations $\{\widehat{q_k}\}$ such that $d_q(\widehat{q_i}, \widehat{q_j}) \approx \widehat{d_p}(\mathbf{p}_i, \mathbf{p}_j) \approx d_q(q_i, q_j)$, with equality if $\widehat{d_p}$ and $\{\widehat{q_k}\}$ are perfectly estimated.

Our approach is similar to [31]. While the authors reconstruct 2D images from 1D projections, they rely on the same two-step approach: they (i) estimate distances as $\widehat{d_p}(\mathbf{p}_i, \mathbf{p}_j) = \|\mathbf{p}_i - \mathbf{p}_j\|_2$ then (ii) recover the orientations by spectrally embedding that distance graph. The Euclidean distance is however not robust to perturbations: for example, two projections that only differ by a shift $\mathbf{S_t}$ of one pixel would be considered far apart while their orientations are the same. They noted that issue and we observed it too (Appendix E). To circumvent this, we propose to *learn* $\widehat{d_p}$ from examples (§2.2).

## 2.1 Representation of orientations with quaternions

The orientation of a 3D particle with respect to the microscope's detector plane is a rotation relative to a reference orientation (Figure 1). The group of all 3D rotations under composition is identified with $\mathbf{SO}(3)$, the group of $3 \times 3$ orthogonal matrices with determinant 1 under matrix multiplication. A rotation matrix $\mathbf{R}_{\boldsymbol{\theta}} \in \mathbf{SO}(3)$ can be decomposed as a product of $\binom{3}{2} = 3$ independent rotations, for example as $\mathbf{R}_{\boldsymbol{\theta}} = \mathbf{R}_{\theta_3} \mathbf{R}_{\theta_2} \mathbf{R}_{\theta_1}$, where $\boldsymbol{\theta} = (\theta_3, \theta_2, \theta_1) \in [0, 2\pi[ \times [0, \pi] \times [0, 2\pi[$ are the (extrinsic and proper) Euler angles in the $ZYZ$ convention (a common parameterization in cryo-EM) [32].

While Euler angles are a concise representation of orientation (3 numbers for 3 degrees of freedom), they suffer from a topological constraint—there is no covering map from the 3-torus to $\mathbf{SO}(3)$—which manifests itself in the *gimbal lock*, the loss of one degree of freedom when $\theta_2 = 0$. This makes their optimization by gradient descent (§2.3) problematic. On the other hand, optimizing rotation matrices (made of 9 numbers) would require computationally costly constraints (orthogonality and determinant 1) to reduce the degrees of freedom to 3. Moreover, the distance between orientations cannot be directly computed from Euler angles and is costly (30 multiplications) to compute from rotation matrices [33]. We solve both problems by representing orientations with unit quaternions.

Quaternions $q \in \mathbb{H}$ are an extension of complex numbers[2] of the form $q = a + b\boldsymbol{i} + c\boldsymbol{j} + d\boldsymbol{k}$ where $a, b, c, d \in \mathbb{R}$. Unit quaternions $q \in \mathbb{S}^3$, where $\mathbb{S}^3 = \{q \in \mathbb{H} : |q| = 1\}$ is the 3-sphere (with the additional group structure inherited from quaternion multiplication), concisely and elegantly represent a rotation of angle $\theta$ about axis $(x_1, x_2, x_3)$ as $q = \cos(\theta/2) + x_1 \sin(\theta/2)\boldsymbol{i} + x_2 \sin(\theta/2)\boldsymbol{j} + x_3 \sin(\theta/2)\boldsymbol{k}$. They parameterize rotation matrices as

$$\mathbf{R}_q = \begin{pmatrix} a^2 + b^2 - c^2 - d^2 & 2bc - 2ad & 2bd + 2ac \\ 2bc + 2ad & a^2 - b^2 + c^2 - d^2 & 2cd - 2ab \\ 2bd - 2ac & 2cd + 2ab & a^2 - b^2 - c^2 + d^2 \end{pmatrix}.$$

---

[1] Up to protein symmetries, which we discuss later.

[2] The algebra $\mathbb{H}$ is similar to the algebra of complex numbers $\mathbb{C}$, with the exception of multiplication being non-commutative.

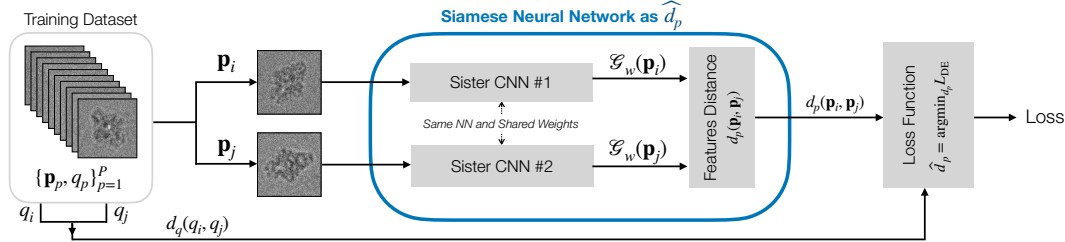

Figure 4: Distance learning. We are looking for a distance $\widehat{d_p}$ between projections that is an accurate estimator of the distance $d_q$ between their orientations. We propose to parameterize $\widehat{d_p}$ as a Siamese neural network (SNN), trained on a synthetic dataset of projections with associated orientation.

Note that $\mathbb{S}^3 \to \mathbf{SO}(3)$ is a two-to-one mapping (a double cover) as $q$ and $-q$ represent the same orientation. Unlike Euler angles, $\mathbb{S}^3$ is isomorphic to the universal cover of $\mathbf{SO}(3)$. Hence, the distance between two orientations, *i.e.*, the length of the geodesic between them on $\mathbf{SO}(3)$, is

$$d_q : \mathbb{S}^3 \times \mathbb{S}^3 \to [0, \pi],$$
$$d_q(q_i, q_j) = 2 \arccos\left(|\langle q_i, q_j \rangle|\right), \tag{2}$$

where $\langle \cdot, \cdot \rangle$ is the inner product, and the absolute value $|\cdot|$ ensures that $d_q(q_i, q_j) = d_q(q_i, -q_j)$. The distance $d_q(q_i, q_j)$ corresponds to the magnitude of the rotation $\mathbf{R}_*$ such that $\mathbf{R}_{q_i} = \mathbf{R}_* \mathbf{R}_{q_j}$ [33].

## 2.2 Distance learning

We aim to estimate a function $\widehat{d_p}$ such that $\widehat{d_p}(\mathbf{p}_i, \mathbf{p}_j) \approx d_q(q_i, q_j)$. While we could in principle design $\widehat{d_p}$, that would be intricate—if not impossible—partly because the invariants are difficult to specify. We instead opt to learn $\widehat{d_p}$, capitalizing on (i) the powerful function approximation capabilities of neural networks, and (ii) the possibility to generate realistic datasets supported by the availability of numerous 3D atomic models[3] and our ability to model the cryo-EM imaging procedure.

From a training dataset $\{\mathbf{p}_i, q_i\}_{i=1}^{P}$, we learn the projection distance

$$\widehat{d_p} = \underset{d_p}{\arg\min}\, L_{\text{DE}}, \quad \text{where} \quad L_{\text{DE}} = \sum_{i,j} \left| d_p(\mathbf{p}_i, \mathbf{p}_j) - d_q(q_i, q_j) \right|^2 \tag{3}$$

is the loss and $d_q$ is defined in (2). The $d_p$ is parameterized as the Siamese neural network (SNN) [34]

$$d_p(\mathbf{p}_i, \mathbf{p}_j) = d_f(\mathcal{G}_w(\mathbf{p}_i), \mathcal{G}_w(\mathbf{p}_j)),$$

where $\mathcal{G}_w$ is a convolutional neural network with weights $w$ that is trained to extract the most relevant features $\mathbf{f}_i \in \mathbb{R}^{n_f}$ from a projection $\mathbf{p}_i$. SNNs, also termed "twin networks", are commonly used in the field of deep metric learning to learn similarity functions [35]. We set the feature space distance $d_f$ as the cosine distance to facilitate the learning of a $\widehat{d_p}$ that respects the elliptic geometry of $\mathbb{S}^3$ (Appendix F). Figure 4 illustrates the proposed learning paradigm.

As evaluating a sum over $P^2$ pairs is computationally intractable for cryo-EM datasets with typically $P$ in the order of $10^5$ projections, we sample the sum and minimize (3) with stochastic gradient descent (SGD) over small batches of pairs. The weights $w$ are updated by back-propagation.

The architecture of $\mathcal{G}_w$ is described in Appendix G. When designing the architecture, we constrain the functional space from which the trained $\mathcal{G}_w$ is drawn and express our prior expert knowledge. For example, we realize shift invariance, *i.e.*, a guarantee that a shift $\mathbf{S}_t$ does not change our estimated distances and orientations, with a fully convolutional architecture. Size invariance, *i.e.*, taking projections $\mathbf{p}$ of varying sizes $n_p$ while yielding a representation $\mathbf{f}$ of a fixed size $n_f$, is realized by a final average pooling layer. As we do not (yet) know how to realize an invariance to noise or PSF, we resort to data augmentation, *i.e.*, training on perturbed projections. In §3.4, we show that a built-in invariance (shift) is far preferable to one learned through augmentation (noise). Finally, as projections

---

[3]https://www.ebi.ac.uk/pdbe/emdb

125 are made by integrating through the 3D volume, projections from opposed directions are mirrors of
126 each other.[4] That is another kind of physical knowledge that should ideally be built into our method.

127 One could hope to train $\mathcal{G}_w$ to directly map projections to orientations as $\widehat{q}_i = \mathbf{f}_i = \mathcal{G}_w(\mathbf{p}_i)$. While
128 that would avoid the orientation recovery step, a space of $n_f = 4$ dimensions does not have room for
129 $\mathcal{G}_w$ to represent the other factors of variation in $\mathbf{p}$, such as different noise levels, PSFs, or proteins.
130 We tested that hypothesis in Appendix F.

## 2.3 Orientation recovery

132 The task of recovering points based on their relative distances has been extensively studied. Many
133 methods aim at mapping high-dimensional data onto a lower-dimensional space while preserving
134 distances, primarily for dimensionality reduction and data visualization. Well-known examples
135 include MDS [36], Isomap [37], LLE [38], Laplacian eigenmaps [39], t-SNE [40], and UMAP [41].
136 The embedding of distance matrices in Euclidean space (given by their eigenvectors) is especially
137 well-described. In particular, the framework of Euclidean distance matrices (EDMs) [42] provides
138 theoretical guarantees on the recovery of points from distances.

139 We however aim to embed the orientations $q$ in $\mathbb{S}^3$ (§2.1), a setting for which we are unaware of any
140 theoretical characterization (*e.g.*, on the shape of the loss function or its behavior when distances are
141 missing or noisy). The fact that $\mathbb{S}^3$ is locally Euclidean does however offer some hope. Indeed, despite
142 the non-convexity and the lack of theoretical guarantees, we are able to appropriately minimize our
143 loss function, as we experimentally demonstrate in Appendix D.

144 We recover the orientations of a set of projections $\left\{\mathbf{p}_k\right\}_{k=1}^{P}$ through

$$\left\{\widehat{q}_k\right\}_{k=1}^{P} = \underset{\{q_k \in \mathbb{S}^3\}}{\arg\min} L_{\mathrm{OR}}, \quad \text{where} \quad L_{\mathrm{OR}} = \sum_{i,j} \left|\widehat{d}_p\left(\mathbf{p}_i, \mathbf{p}_j\right) - d_q\left(q_i, q_j\right)\right|^2 \tag{4}$$

145 is the loss and $\widehat{d}_p$ is the estimator trained in (3). Note that the sole difference with (3) is that the
146 minimization is performed over the orientations $q$ rather than the distance $d_p$. Here again, we sample
147 the sum in practice and minimize (4) with mini-batch SGD. Sampling the sum amounts to building a
148 sparse (instead of complete) distance graph before embedding, a common strategy.

## 2.4 Evaluation

150 While not a part of the method *per se*, we must evaluate the quality of the recovered orientations.
151 Unfortunately, we cannot directly take the difference between the recovered orientations $\{\widehat{q}_k\}_{k=1}^{P}$
152 and the true orientations $\{q_k\}_{k=1}^{P}$ as orientations are rotations up to an arbitrary reference orientation.
153 Any global rotation or reflection of the recovered orientations is as valid as any other, *i.e.*, $d_q(q_i, q_j) =$
154 $d_q(\mathbf{T}q_i, \mathbf{T}q_j) \,\forall\, \mathbf{T} \in \mathbf{O}(4)$, where $\mathbf{O}(4)$ is the group of $4 \times 4$ orthogonal matrices. Hence, we align
155 the sets of orientations and compute the *mean orientation recovery error* as

$$E_{\mathrm{OR}} = \min_{\mathbf{T} \in \mathbf{O}(4)} \frac{1}{P} \sum_{i=1}^{P} \left|d_q\left(q_i, \mathbf{T}\widehat{q}_i\right)\right|. \tag{5}$$

156 We implement $\mathbf{T}$ as a product of $\binom{4}{2} = 6$ independent rotations and an optional reflection:

$$\mathbf{T} = \begin{bmatrix} m & \mathbf{0} \\ \mathbf{0} & \mathbf{I} \end{bmatrix} \prod_{1 \le i < j \le 4} \mathbf{T}_{\theta_{ij}}, \quad m \in \{-1, 1\},\ \theta_{ij} \in [0, 2\pi[, \tag{6}$$

157 where $\mathbf{T}_{\theta_{ij}} \in \mathbf{SO}(4)$ is a rotation by angle $\theta_{ij}$ on the $(x_i, x_j)$ plane.

158 In practice, we again minimize (5) with mini-batch SGD. Because $\mathbf{O}(4)$ is disconnected, we optimize
159 the 6 angles separately for $m = 1$ (proper rotations) and $m = -1$ (improper rotations). Figure 15
160 shows an alignment to $E_{\mathrm{OR}} = 0$ after a perfect recovery.

---

[4]That fact prevents the resolution of chirality, *i.e.*, we cannot distinguish a protein from its mirrored form.

## 3 Experiments

We first evaluated whether orientation recovery through (4) was feasible assuming perfect distances, and how it was affected by errors in the distances (§3.2). We then learned to estimate the distances through (3), and evaluated the accuracy of this procedure (§3.3) and its robustness to perturbations of the projections (§3.4). Finally, we ran the whole machinery on a synthetic dataset to assess how well orientations could be recovered from estimated distances (§3.5).

### 3.1 Experimental conditions

**Density maps.** We considered two proteins (Figure 10): the $\beta$-galactosidase, a protein with a dihedral (D2) symmetry, and the lambda excision HJ intermediate (HJI), an asymmetric protein with local cyclic (C1) symmetry. Their deposited PDB atomic models are 5a1a [43] and 5j0n [44], respectively. From these atomic models, we generated the density maps in Chimera [45] by fitting the models with a 1Å map for 5a1a and a 3.67Å map for 5j0n; this gave us a volume of $110 \times 155 \times 199$ voxels for 5a1a and one of $69 \times 57 \times 75$ voxels for 5j0n.

**Protein symmetries.** Symmetries are problematic when learning distances: two projections can be identical while not originating from the same orientation, which breaks an axiom of distance functions (identity of indiscernibles). Figure 16b illustrates this problem. To capture only one of four identical projections of 5a1a, we restricted directions to $(\theta_2, \theta_1) \in [0, \pi[ \times [0, \frac{\pi}{2}[$ (a quarter of the sphere, illustrated in Figure 12a) for that protein. This treatment of symmetries is incomplete[5] but sufficient for a proof-of-concept.

**Projections.** Using the ASTRA projector [46], we generated $P = 5,000$ synthetic projections of $275 \times 275$ pixels (downsampled to $116 \times 116$) for 5a1a and $116 \times 116$ pixels for 5j0n, taken from uniformly sampled orientations.[6] We then perturbed the measurements with different levels of additive Gaussian noise [47, 48] and off-centering shifts. Figure 11 displays some samples.

**Datasets.** For each protein, we split the projections into training, validation, and test subsets, and created *disjoint* pairs of projections from each (Table 1). The training and validation sets were used to train and evaluate the SNN, while the test set was used to evaluate orientation recovery given a trained SNN. Sampling orientations (mostly) uniformly induces a distribution of distances that is skewed towards larger distances (shown in Figure 12b). As this would skew $L_{\mathrm{DE}}$ and bias $\widehat{d}_p$, we further sampled $1\%$ of the training and validation pairs to make the distribution of distances uniform—for $\widehat{d}_p$ to be uniformly accurate over the whole $[0, \pi]$ range of distances (see Appendix B for further illustrations). While $1,650$ projections were enough to perfectly reconstruct the density maps (as shown in Figures 9e and 9j), our method is not limited by the number of projections as optimization is done per batch. Optimization settings are described in Appendix C.

### 3.2 Sensitivity of orientation recovery to errors in distance estimation

We first evaluated the feasibility of orientation recovery assuming that the exact distances were known. The method successfully recovers the orientation of every projection in this case (see Appendix D).

To evaluate the robustness of (4), we perturbed the distances prior to recovery with an error sampled from a Gaussian distribution with mean 0 and variances $\sigma^2 \in [0.0, 0.8]$. Figure 5 shows that the recovery error $E_{\mathrm{OR}}$ is a monotonic function of the error in distances: from $E_{\mathrm{OR}} = 0$ with exact distances to $E_{\mathrm{OR}} \approx 0.2$ radians ($\approx 11.5°$) for $\sigma^2 = 0.8$.

These results demonstrate that the performance of orientation recovery (4) depends on the quality of the estimated distances, which advocates for a proper and extensive training of the SNN. Moreover, we observe that $L_{\mathrm{OR}}$ is a reliable proxy for $E_{\mathrm{OR}}$, allowing us to assess recovery performance in the absence of ground-truth orientations (*i.e.*, when recovering the orientations of real projections).

---

[5]The remaining issue is that one of four distances is arbitrarily chosen per pair of projections.

[6]Orientations used in §3.2 (Figure 5) and §3.4 (Figure 7) were actually obtained by uniformly sampling the Euler angles $\boldsymbol{\theta}$, constrained to $(\theta_3, \theta_2, \theta_1) \in [0, 2\pi[ \times [0, \frac{\pi}{2}[ \times [0, 2\pi[$ for 5j0n. Our conclusions would be identical if orientations were uniformly sampled from $\mathbf{SO}(3)$ instead.

Table 1: Split of $P = 5,000$ projections in training, validation, and test subsets.

| Dataset | $P$ | $P^2$ | Used pairs |
|---|---|---|---|
| Training | 2,512 (50%) | 6,310,144 | 63,101 |
| Validation | 838 (17%) | 702,244 | 7,022 |
| Test | 1,650 (33%) | 2,722,500 | 2,722,500 |

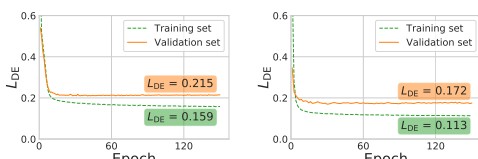

(a) Loss converged on `5j0n` (left) and `5a1a` (right).

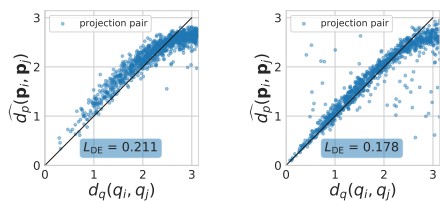

(b) Relationship between $\widehat{d}_p$ and $d_q$ on $1,000$ pairs from the test sets of `5j0n` (left) and `5a1a` (right).

Figure 6: Distance learning.

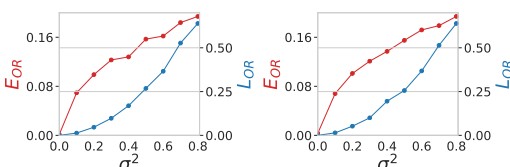

Figure 5: Orientation recovery from perturbed distances on `5j0n` (left) and `5a1a` (right).

## 3.3 Learning to estimate distances

We evaluated the ability of the SNN to learn to approximate the orientation distance $d_q$. For comparison, we evaluated a baseline, the Euclidean distance $\widehat{d}_p(\mathbf{p}_i, \mathbf{p}_j) = \|\mathbf{p}_i, \mathbf{p}_j\|_2$, in Appendix E.

Figure 6a shows the convergence of $L_{\mathrm{DE}}$, reached in about 50 epochs. Figure 6b shows the relationship between the distance $\widehat{d}_p$ estimated from projections and the true distance $d_q$. The outliers for `5a1a` are explained by our incomplete treatment of its symmetry. While our learned distance function is a much better estimator than the Euclidean distance—compare Figure 6b with Figure 16—they share one characteristic: both plateau and underestimate the largest distances. We did attenuate the phenomenon by sampling training distances uniformly (see §3.1), and the issue is much less severe than with the Euclidean distance. An alternative could be to only rely on smaller distances for recovery. That would however require the addition of a spreading term in (4) to prevent the recovered orientations to collapse.

These results confirm that a SNN is able to estimate differences in orientations from projections alone, even though much has yet to be gained from improving upon the rather primitive SNN architecture we are currently using. The use of additional training data should help further diminish overfitting.

## 3.4 Sensitivity of distance learning to perturbations in the projections

We first demonstrated that the learning of distances is insensible to off-centering shifts (Figure 7a), which is expected given that shift invariance is built in our SNN (see §2.2).

As we cannot—or do not yet know how to—build noise invariance in the SNN architecture, we trained the SNN on noisy projections and evaluated whether it could learn to treat noise as an irrelevant information. Figure 7b shows $E_{\mathrm{OR}} \approx 0.16$ radians ($\approx 9°$) for noiseless projections and $E_{\mathrm{OR}} \approx 0.42$ radians ($\approx 24°$) for a more realistic noise variance of $\sigma^2 = 16$ (with signal-to-noise ratio of -12 dB). Whereas a naive distance function (*e.g.*, an Euclidean distance) would be extremely sensitive to noise, the SNN mostly learned to discard it. Moreover, the observed overfitting indicates that more training data should further decrease the sensitivity of the SNN to noise.

Note that we did not evaluate sensitivity to the PSF at this stage but expect a similar behavior.

Here again (§3.2), we observed that (i) the estimation of more accurate distances (a smaller $L_{\mathrm{DE}}$) leads to the recovery of more accurate orientations (a smaller $L_{\mathrm{OR}}$ and $E_{\mathrm{OR}}$), and that (ii) an higher recovery loss $L_{\mathrm{OR}}$ induces an higher error $E_{\mathrm{OR}}$.

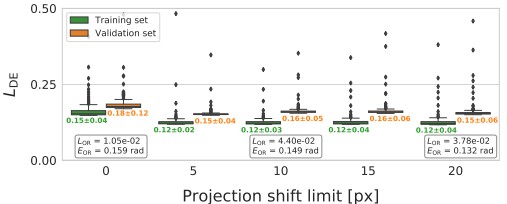
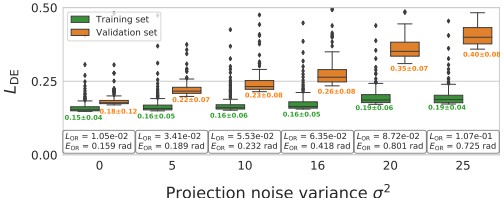

(a) Learning from shifted projections $\{\mathbf{S}_{\mathbf{t}_i}\mathbf{P}_{\boldsymbol{\theta}_i}\mathbf{x}\}$, with shifts $t_{i_1}$ and $t_{i_2}$ sampled from a triangular distribution with mean 0 and of increasing limits.

(b) Learning from noisy projections $\{\mathbf{P}_{\boldsymbol{\theta}_i}\mathbf{x} + \mathbf{n}\}$, with white noise $\mathbf{n} \sim \mathcal{N}(0, \sigma^2\mathbf{I})$ of increasing variance $\sigma^2$.

Figure 7: Sensitivity of distance learning to perturbations in the projections of 5j0n. The box plots show the distance learning loss $L_{\mathrm{DE}}$ (the distribution is taken over epochs). Boxes show the orientation recovery loss $L_{\mathrm{OR}}$ and error $E_{\mathrm{OR}}$.

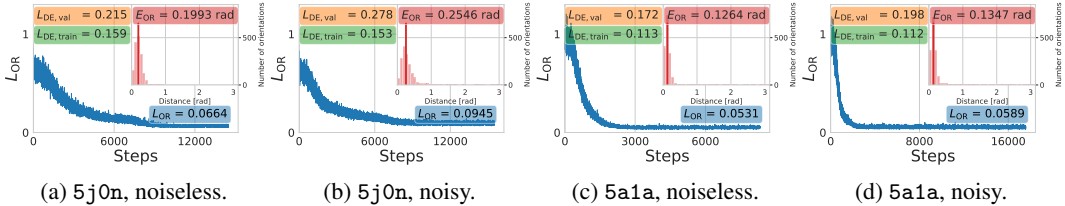

(a) 5j0n, noiseless.     (b) 5j0n, noisy.     (c) 5a1a, noiseless.     (d) 5a1a, noisy.

Figure 8: Distance learning and orientation recovery from estimated distances. The green and orange boxes show $L_{\mathrm{DE}}$ (3) on the training and validation sets. The blue curve shows the evolution of the recovery loss until convergence, with the minimum $L_{\mathrm{OR}}$ (4) highlighted. The red histogram shows the errors in the recovered orientations $\{d_q(q_i, \mathbf{T}\widehat{q}_i)\}$, with the mean $E_{\mathrm{OR}}$ (5) highlighted.

## 3.5 Orientation recovery and reconstruction of density maps

As a proof-of-concept, we attempted to solve the full inverse problem posed by (1), *i.e.*, to reconstruct the density maps $\widehat{\mathbf{x}}$ from sets of projections $\{\mathbf{p}_i\}$ and their orientations $\{\widehat{q}_i\}$ recovered through the proposed method. It is worth noting that, at this stage of development, we only trained the SNN on projections originating from the protein we were attempting to reconstruct. In addition, reconstruction was performed with a direct reconstruction algorithm (ASTRA's GPU implementation of the CGLS algorithm) rather than with a robuster iterative method. This is a specific experimental case that only partially shines light on the applicability of the method in real situations; this is discussed in §4.

Figure 8a shows the recovery of orientations from distances that were estimated from noiseless projections of 5j0n. A mean error of $E_{\mathrm{OR}} \approx 0.20$ radians ($\approx 11°$) in the recovered orientations led to a reconstruction with a resolution of 12.2Å at a Fourier shell coefficient (FSC) of $0.5$, shown in Figure 9c. As predicted by our other experiments, corrupting the projections with noise ($\sigma^2 = 16$) negatively impacts the quality of the recovered orientations (Figure 8b); the obtained mean error is then $E_{\mathrm{OR}} \approx 0.25$ radians ($\approx 14°$). Unsurprisingly, this leads to a reconstruction with a lower resolution of 15.2Å, shown in Figure 9d. (Note that reconstruction was here obtained from the noiseless projections, the goal being to evaluate only the impact of orientation mis-estimation.)

Finally, Figures 8c,d show the recovery of orientations from noiseless and noisy projections of 5a1a. A mean error of $E_{\mathrm{OR}} \approx 0.13$ radians ($\approx 7°$) in both cases led to reconstructions with resolutions of 8.0Å and 9.6Å, shown in Figures 9h,i. Distance estimation, orientation recovery, and reconstruction performed better on 5a1a than 5j0n because its ground-truth density is of higher resolution.

These results tend to indicate that a reasonable first structure can be reconstructed from projections whose orientations have been recovered through our method.

## 4 Discussion

In this work, we explored the use of distance learning between pairs of 2D cryo-EM projections from a 3D protein structure to infer the unknown orientation at which each projection was imaged

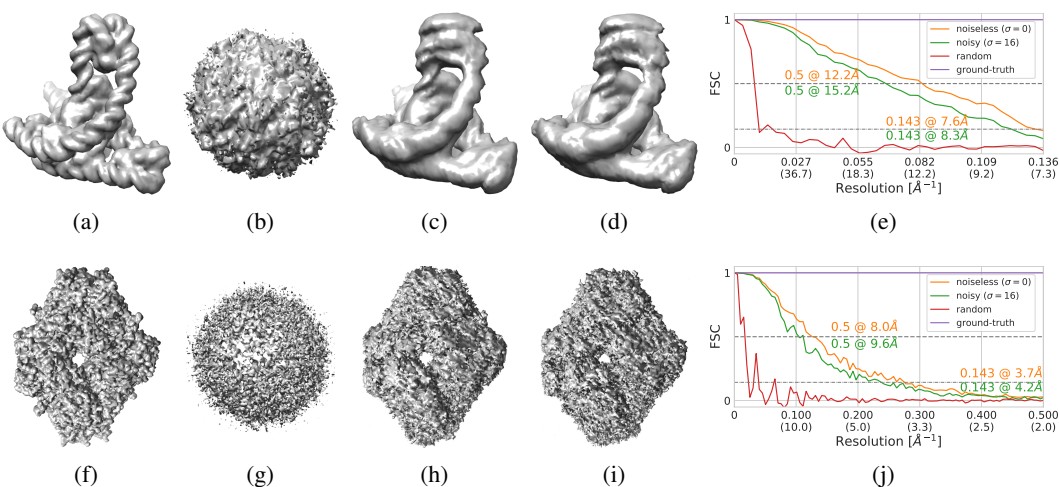

Figure 9: Density maps $\hat{\mathbf{x}}$ reconstructed from (a,f) ground-truth orientations, (b,g) random orientations, (c,h) orientations recovered from noiseless projections, and (d,i) orientations recovered from noisy projections. The Fourier shell correlation (FSC) curves in (e,j) indicate the resolutions of the densities (w.r.t. ground-truth densities, shown in Figures 10b,d).

from. Our two-step method relies on the estimation of pairwise distances between unseen projections, followed by the recovery of the orientations from these distances.

The method has been evaluated on synthetic datasets for two different proteins. The results provide key insights on the viability of the proposed scheme. First, they demonstrate that a SNN can learn a distance function between projections that estimates the difference in their orientation (§3.3) and that is invariant to shifts and robust to increasing levels of noise (§3.4)—an important condition in cryo-EM. Second, they demonstrate that an accurate estimation of distances leads to an accurate recovery of orientations (§3.2, §3.4). Finally, our method was able to recover orientations with an error of $0.12$ to $0.25$ radians (7 to $14°$)—leading to an initial volume with a resolution of $8$ to $15$Å (§3.5). In summary, the more accurate the estimated distances, the more precise the recovered orientations, and, ultimately, the higher-resolution the reconstructed volume.

While the method is not yet ready to be deployed in practice, we believe that a series of developments could make it relevant for single-particle cryo-EM reconstruction.[7] As previously discussed, the results underline the importance of learning an accurate distance estimator. In this regard, the performance of the SNN could be improved. First, the architecture of the twin convolutional neural networks should be expanded and tuned. Second, training could be improved, perhaps by providing more supervision by separately predicting the differences in direction $(\theta_2, \theta_1)$ and in-plane angle $\theta_3$.

Importantly, the SNN would be better trained on a more diverse cryo-EM dataset. Indeed, its success as a faithful estimator eventually relies on our capacity to generate a synthetic training dataset whose data distribution is diverse enough to cover that of unseen projection datasets. Such realistic cryo-EM projections could be generated by relying on a more expressive formulation of the cryo-EM physics and taking advantage of the thousands of atomic models available in the PDB. In particular, a necessary extension will be to include the effects of the PSF and to evaluate its impact.

A final phase of tests before deploying the method on real cryo-EM measurements will be to extensively test the method on "unseen proteins", *i.e.*, proteins whose simulated projections have never been seen by the SNN. In this regard, an interesting aspect of our method is that the twin networks within the SNN intrinsically predict the *relationship* between projections, allowing the SNN as a whole to abstract the particular volume. Learning should benefit from the profound structural similarity shared by proteins—after all, they are all derived from the same 21 building blocks.

Training our 4.5M parameter model (see Appendices G and C) has the following negative environmental impact: it consumes 13 kWh of energy, which produces 6.36 lbs of $CO_2$ on average [49].

---

[7]Note that the present project will not be further continued by its authors due to other professional occupations. Hence, we strongly encourage anyone interested to build on these ideas and, hopefully, make it a practical tool.

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
