# OpenReview forum: "Learning to recover orientations from projections in single-particle cryo-EM"
_NeurIPS.cc/2021/Conference — NeurIPS 2021 Submitted_

### Official Review · Reviewer_RNam · 2021-07-15

**Rating:** 5
**Confidence:** 5

**Summary:**

This work presents a method for estimating the viewing directions of projection images in single-particle cryo-electron microscopy (cryo-EM). The method consists of two steps, first the distance between the viewing angles are estimated using a neural network, then these distance estimates are combined to yield a viewing angle estimate for each projection image. The method is trained and evaluated on synthetic projection images, where it is shown to perform well in producing reconstruction distance estimates are combined to yield a viewing angle estimate for each projection image. The method is trained and evaluated on synthetic projection images, where it is shown to perform well in producing reconstructions of moderate resolution.


**Ethical Concerns:**

No ethical issues are raised by the manuscript.


**Limitations And Societal Impact:**

The authors discuss the limitations of their model and their numerical validation at the end of the manuscript. When it comes to the validation, I think more can be done (as stated in the main part of the review). This being an application of data-driven methods to a problem in structural biology, I do not see any potential negative societal impacts that would emerge directly from this work. The authors are, however, helpful in providing an estimated GHG budget for training their model.


**Main Review:**

This paper presents an interesting method for ab initio reconstruction in single-particle cryo-EM, a difficult problem that stands to benefit from data-driven approaches due to the high noise content of the data, which is otherwise quite structured. The proposed method is promising, but numerical validation in the present work is lacking, especially with respect to comparisons with existing methods or experimental data. Instead, the performance of the method is evaluated in a synthetic-data setting without comparison to other methods and with a network that is trained on the same molecule. It is therefore difficult to judge the wider applicability of the method, despite its inherent appeal. If this type of validation could be provided (even in partial form), I would recommend this paper be accepted as part of the proceedings, but otherwise I ask that it not be included. More detailed comments follow.

As stated above, the problematic component of the work concerns the numerical validation. The motivation and description of the method is quite sound, but could be better explained in parts. For example, the authors switch back and forth between Euler angles θ_i and quaternions q_i (they are simply introduced without explanation on line 63), which can make for confusing reading. The authors also state that the mapping from S³ to SO(3) is a double cover, but do not explain whether this is a problem or not for the problem setting. On lines 120–121, the authors also discuss size invariance, but it is not clear why this is necessary. When are projections expected to be of different sizes? This should not be the case in the numerical experiments discussed since the training data and testing data arise from the same protein structure. A similarly confusing statement is on line 127, where the authors claim that “a space of n_f = 4 dimensions does not have room for G_w to represent other factors of variation”. What does this mean? Why are we constrained to have n_f = 4?

Constructing viewing angle estimates from a set of distance estimates also deserves some closer discussion in Section 2.3. Indeed, constructing a global set of angle estimates from smaller local distance estimates can lead to inconsistencies, as pointed out by Zhao and Singer, 2014 (Section 2.1). Why do we expect that constructing small subsets of the embedding at a time will result in a globally consistent embedding?

For the evaluation method described in Section 2.4, it is a bit problematic to use a non-deterministic error measure since it induces its own “error” for each run. It would be better if this could be replaced with a deterministic error. One way to achieve this is to replace the L¹-type norm used on the rotations and instead use a Frobenius norm on the rotation matrices. In this case, the optimal rotational alignment between the sets can be calculated by an SVD. That being said, perhaps the authors have other reasons for choosing the error measure that make it more appropriate than the simpler Frobenius norm. If that is the case, it should be motivated in the text.

For the numerical validation, it is striking that only uniform distributions of viewing angles are tested. First, these are relatively uncommon in experimental data, with a preferred orientation being the norm. Second, while many algorithms tend to work well for uniform distributions, that is not necessarily the case with non-uniform distributions. It would therefore be a good test of the approach to evaluate its performance in this regime.

A big problem with the evaluation is the choice of datasets. The authors acknowledge this in the discussion, but all the same, it would be good to include the results of some preliminary experiments here. For example, the networks are trained on one of two molecules. What happens if one network is used to predict the distances for the other dataset? Does performance suffer greatly or is there a degree of robustness here. What if the network is trained on both datasets? The universality of the approach will be crucial when extending the method to experimental data, so it is important to know what can be expected here, even at a preliminary level. Other questions related to training mismatch concern noise levels. To what extent can we train on one noise level and test on another?

When it comes to the noise levels, it is hard to extract a useful context. For example, the authors give the noise variance of the data, but without knowing the variance of the images, it is not obvious if this is a high or low noise level. Later in the text, we see that σ² = 16 corresponds to an SNR of –12 dB (which presumably means an SNR of 10 ^ (-1.2), but we are not given the formula). This is a low SNR, to be sure, but not exceptionally low for experimental cryo-EM data. Judging from the images shown in Figure 11 (which presumably have σ² = 16), these are indeed of moderate SNR for cryo-EM.

Another piece of the validation section that is missing is comparison to other methods. In the introduction, a host of alternative methods for ab initio structure determination are listed, but none are applied to the synthetic data used to evaluate the proposed method. It would be simple enough to run the synthetic data through the ab initio pipeline of Cryosparc, Relion, or Aspire to provide a simple baseline for comparison. Right now, the viewing angle accuracies and reconstruction resolutions are hard to place since there is no basis for comparison. This is especially important since it would provide an indication of the benefit derived from a data-driven approach, where specific features in the images are used to estimate the distances. Due to the training–testing regime, it is expected that the proposed method would perform better in this setting (since it possesses a much stronger prior compared to the baseline methods), but this needs to be validated numerically.

Some minor comments:

– Constant references to figures outside of the main text makes for tedious reading. I would try to include these in the main text (perhaps by cutting other figures) or remove references entirely. If this cannot be resolved, I suggest labeling these figures so that it is clear that they refer to supplementary material (Figure S1, etc.).

– On line 100, it should be clarified what is meant by the “magnitude” of a rotation.

– On line 102, what does it mean the “design” the estimator, and why is it intricate or impossible? What do invariants have to do with this?

– On line 120, a CNN provides equivariance (covariance) not necessarily invariance to translation (unless a pooling layer is included).

– On line 143, “our loss function” is referred to but has yet to be introduced.

– On lines 172, 244, 248, 252, and 268, spaces are missing between the number and the unit (Å).

– On lines 180, 191, and the caption of Table 1, there is an extra space after the comma in the numbers larger than one thousand.

– In footnote 6, sampling the Euler angles from uniform distributions does not yield a uniform distribution on SO(3). The colatitude (θ₂) has to be reweighted.

– On line 244, “an” should be removed.

– On lines 228–229, what does it mean that “the observed overfitting indicates that more training data should further decrease the sensitivity of the SNN to noise”?

– On line 240, “robuster” should be “more robust”.

– On line 253, why does higher resolution of the ground truth imply better reconstruction resolution?

– Several article and journal titles are improperly capitalized in the bibliography, including for references [11], [12], [18], [20], [21], [28], [30], [31], [32], [40], [41], and [43].


**Time Spent Reviewing:**

2

---

> ### Author Response · Authors · 2021-08-09
> **Rebuttal to Review by Reviewer RNam**
>
> Thank you for your time and thoughtful comments.
>
> Regarding the request to further evaluate the method on unseen proteins and to compare it to existing pipelines: We refer the reviewer to our top-level "General Rebuttal" comment, where those questions are addressed.
>
> We switch back and forth between Euler angles θ_i and quaternions q_i as they are equivalent representations of the same information. We do however understand this may sometimes be confusing to readers; this could be addressed in a revised version of the manuscript.
>
> The fact that the mapping from S³ to SO(3) is a double cover was meant as an explanation for equation (2). We agree it could have been better conveyed and will update line 99 to read "The absolute value $\left| \cdot \right|$ ensures that $d_q(q_i, q_j) = d_q(q_i, -q_j)$ as $q$ and $-q$ represent the same orientation because $\mathbb{S}^3 \rightarrow \mathrm{\mathbf{SO}}(3)$ is a two-to-one mapping (a double cover)."
>
> "When are projections expected to be of different sizes?" This could happen with real data, though that is indeed not the case with the current experiments.
>
> The number of features $n_f$ can be freely chosen, and was set to $n_f = 512$ in our experiments. On line 127, we meant that a feature space made of only $n_f = 4$ dimensions (necessary if one would hope to predict the orientations directly) is too constraining (demonstrated in Appendix F). We agree that this paragraph might confuse readers and will only be pointing to Appendix F for experiments on the number of features $n_f$ and move the discussion there.
>
> Zhao and Singer, 2014 are interested in estimating a distance between viewing directions (θ_2, θ_1). In Section 2.1, they argue that any method that requires images to be aligned with respect to in-plane rotations θ_3 is bound to fail because the hairy ball theorem prevents such an alignment to be globally consistent. We agree with the argument but fail to see how that relates to our method: We don't try to align on θ_3 then estimate a distance between viewing directions (θ_2, θ_1). We estimate a distance between whole orientations (θ_3, θ_2, θ_1).
>
> We chose the mean orientation recovery error because it is intuitive and simple to interpret. Despite that error being non-deterministic, the evaluation of (5) was easily reproduced in practice. But we do agree that using the Frobenius norm on the rotation matrices could be an interesting alternative choice.
>
> We did evaluate our method on non-uniformly distributed viewing angles in Appendix B. Performance was barely affected.
>
> As we do not know (yet) how to build NNs that are invariant to noise, we need to resort to data augmentation. To generalize to any noise level, the NN should thus be trained on a variety of noise levels. We are confident that the NN would generalize well as our experiments show it was mostly indifferent to noise. That said, transfer between noise levels, PSFs, and proteins should be part of a comparison with established pipelines, which we argue in our top-level "General Rebuttal" comment should be addressed in a separate contribution.
>
> The clean images have variance 1 and the SNR formula you assumed is correct; we will add this information to the manuscript.
>
> We do agree that the level of noise currently used in our experiments, although already significant, does not fully cover the most severe cases of degradation observed in cryo-EM. That being said, we believe that testing our method on synthetic measurements with a SNR of -12 dB is a legit first step in demonstrating its potential in real situations. Note that Figure 7b shows the performance for a σ² varying between 0 and 25, where the upper limit corresponds to a SNR of -14 dB.
>
> We will address the minor comments in the revised manuscript and answer the questions below.
> * Line 102: The estimator $\widehat{d_p}$ is a function that could in principle be "designed by hand" or chosen among known functions (e.g., the Euclidean distance $\widehat{d_p}(\mathbf{p}_i, \mathbf{p}_j) = \| \mathbf{p}_i - \mathbf{p}_j \|_2$ shown in Appendix E) instead of learned from data. We want that function to be invariant to irrelevant transformations (e.g., shift, noise, PSF). As we do not know how to design such a function by hand, we resort to learn it from examples.
> * Footnote 6: Some experiments were done with a uniform distribution over SO(3), while others (§3.2 and §3.4) were done with a uniform distribution over Euler angles. We empirically verified that sampling uniformly or non-uniformly over SO(3) barely makes a difference in Appendix B.
> * Lines 228–229: The SNN is overfitting, a sign that it was not trained on enough data. More data should undoubtedly help it generalize better.
> * Line 253: The more detailed the projections, the easier it can become to distinguish between two closely-related projections (i.e., with close orientations) for an identical level of noise. From a more general standpoint, the higher the resolution of the projections, the higher the resolution of the reconstruction for a same accuracy of orientation recovery.

---

### Official Review · Reviewer_Z4BJ · 2021-07-16

**Rating:** 6
**Confidence:** 4

**Summary:**

In this paper, “Learning to recover orientations from projections in single-particle cryo-EM,” the authors present an approach to recovering unknown projection orientations based on learning to predict the distance between projection angles given observed projections. Given these predicted projection angle distances, it is possible to recover the orientations themselves by solving for the orientations with distances that best match the predicted distances. The distances themselves are predicted using a Siamese neural network that is trained on projection pairs with known orientation distances. The authors demonstrate this approach on two synthetic datasets.

**Ethical Concerns:**

None.

**Limitations And Societal Impact:**

Limitations and societal impact are sufficiently addressed.

**Main Review:**

Although the method has some clear limitations and is only demonstrated on synthetic datasets, this paper presents a compelling and well described method and proof-of-concept demonstration. The paper is well written, and the problem and method are well motivated with relevant background. Specific comments follow below.
1.	The big question: how will this work in practice on real cryoEM datasets where orientations are not known? Even training this on real cryoEM datasets is potentially problematic because the orientations of the particles were estimated to produce the reconstruction. Will a network be able to generalize over particles from different structures?
2.	What happens when the particles are not derived from single rigid structure (i.e., there is conformational heterogeneity)?
3.	When minimizing (4) to solve for the orientations from the distances, how are the orientations initialized? Similarly to t-SNE and other distance-based embedding methods, it seems like the objective is non-convex and so orientations may be mis-estimated due to poor initialization. How quickly does the minimization converge?
4.	It would be useful to see a comparison with orientations estimated using a common lines approach (e.g., http://spr.math.princeton.edu/orientation). It would also be useful to see the orientations estimated by an iterative reconstruction procedure as baselines to better understand how well the orientations in these datasets can be estimated by other approaches.
Overall, I think there are some nice ideas here that are well motivated and described. The main weaknesses are the question about how this will apply to real cryoEM data and the lack of strong baselines for orientation estimation.

**Time Spent Reviewing:**

3

---

> ### Author Response · Authors · 2021-08-09
> **Rebuttal to Review by Reviewer Z4BJ**
>
> Thank you for your time and thoughtful comments.
>
> Regarding your questions:
> 1. The extension of the method to handle real cryo-EM datasets with unknown orientations is discussed in Section 4: "The success [of the method] as a faithful estimator eventually relies on our capacity to generate a synthetic training dataset whose data distribution is diverse enough to cover that of unseen projection datasets. Such realistic cryo-EM projections could be generated by relying on a more expressive formulation of the cryo-EM physics and taking advantage of the thousands of atomic models available in the PDB." In other words, our hope is that by simulating a large and diverse training dataset whose statistics sufficiently approach that of real cryo-EM datasets, the SNN will generalize to real proteins. We also refer the reviewer to our top-level "General Rebuttal" comment, where this question is addressed from a higher-level perspective.
> 2. This is indeed a critical point, which would deserve to be further discussed in a revised version of this paper. A pragmatic solution at this stage would be to rely on the 2D classification methods of existing cryo-EM packages, which try to untangle the different configurations within a cryo-EM projection dataset by classifying the 2D projections based on certain similarity criteria. Once the various configurations have been "separated", our method could then be run independently on each data subset.
> 3. The orientations are randomly initialized, drawn from a uniform distribution over Euler angles. The objective function is indeed non-convex, though we found it to almost always converge to the same solution (up to a global rotation). We believe that initialization is not much of a problem here, as the embedding space is the "proper" space of 3D rotations. On the other hand, methods like t-SNE embed in a low-dimensional Euclidean space that cannot faithfully represent the data. The convergence time is reported in Appendix C and is ~3.75 hours (though Figure 8 shows it reaches a plateau 1-2 hours earlier).
> 4. Here as well, we refer the reviewer to our top-level "General Rebuttal" comment, where the general question of the comparison with existing methods (which could indeed include a common-lines approach) is discussed.

---

### Official Review · Reviewer_9yDH · 2021-07-16

**Rating:** 6
**Confidence:** 4

**Summary:**

This paper uses deep learning to recover the orientations of molecules measured in the context of single particle cyroEM. The proposed method first estimates the pairwise distances between the angles of molecules based on the their projections. This is accomplished by embedding the projections into a feature space that was trained such that the cosine similarity between two features is equal to the distance between the orientations of the two molecules. Next, the authors solved an optimization problem (3) to recover the orientations, up to global ambiguities, from the pairwise distances.


**Limitations And Societal Impact:**

Authors are transparent about the limitations of current method. Single particle cryoEM is an important problem.


**Main Review:**


# Originality
Recovering cryoEM orientations or structure with DL is not new, but the previous works I'm familiar with have mostly focused on solving the task with dimensionality reduction. I haven't seen a paper tackle the problem by learning to estimate the pairwise distances between angles.

# Quality
The paper did a good job of validating each proposed contribution in isolation. Simulations results were overall thorough, though there was no real experimental data.

# Clarity
Well-written

# Significance
It's not evident the proposed method will work with real data and the method was trained on the same molecules whose orientations it was going to reconstruct.

It's also unclear if the proposed methods improvements will actually improve the final quality of cryoEM reconstructions. This tasks performed in this paper are essentially preprocessing steps that recover an initialization that is fed into another algorithm (e.g., expectation maximization and RELION) that would reconstruct the structure. It's unclear if the improvements in the init help the end result.

# Related work
Recent work in DL for single particle cryoEM has attempted to recover the 3D structure without explicitly recovering the angles using adversarial networks [A]. It may be worth mentioning.
[A] Gupta, Harshit, et al. "Cryogan: A new reconstruction paradigm for single-particle cryo-em via deep adversarial learning." BioRxiv (2020).

# Questions
It's my understanding that some molecules will preferentially align in certain directions, so that the view angles in cyroEM are not uniformly distributed. This is a major issue for embedding-based methods. How would non-uniformly distributed angles effect the proposed method?



**Time Spent Reviewing:**

2.5

---

> ### Author Response · Authors · 2021-08-09
> **Rebuttal to Review by Reviewer 9yDH**
>
> Thank you for your time and thoughtful comments.
>
> Regarding the request to further evaluate the method on unseen proteins and to compare it to existing pipelines: We refer the reviewer to our top-level "General Rebuttal" comment, where those questions are addressed.
>
> On how an improvement of the initial angle estimation translates into an improvement of the reconstruction: It has been shown in various works [R1, R2] that the outcome of iterative refinement procedures in cryo-EM is predicated on the quality of the initial reconstruction, or, equivalently, on the initial estimation of the orientations.
>
> [R1] Carlos Oscar Sanchez Sorzano et al., "Optimization problems in electron microscopy of single particles", Annals of Operations Research, vol. 148,no. 1, pp. 133–165, 2006
>
> [R2] Richard Henderson et al., "Outcome of the first electron microscopy validation task force meeting", Structure, vol. 20, no. 2, pp. 205–214, 2012
>
> We will add the suggested reference to [A] in the revised manuscript.
>
> On how non-uniformly distributed angles would affect the proposed method: We tried it in Appendix B (with a uniform sampling of Euler angles, which is non-uniform on SO(3), see Figure 12) and performance was barely affected (see Figure 13).

---

### Official Review · Reviewer_zuH5 · 2021-07-21

**Rating:** 4
**Confidence:** 4

**Summary:**

The authors tackle the problem of image orientation estimation in single particle cryo-electron microscopy. This imaging technique generates a dataset of ~10^{5-7} 2D projection images of a 3D protein with unknown pose (elements of SO(3)xR2). Once poses are inferred, the 3D structure may be reconstructed with standard tomographic projection techniques. To estimate the distances, the authors propose a Siamese convolutional network architecture, which takes as input a pair of images, and attempts to predict the distance between their orientations (parameterized as quaternions). Once this distance function is learned (from a training set of posed images on a single protein), gradient-based optimization is used to infer each image’s pose. The authors show results on two small, synthetic datasets of 5000 images.

**Limitations And Societal Impact:**

Yes

**Main Review:**

I appreciate the detailed study performed by the authors, however I do not think this paper is suitable for publication at NeurIPS as it focuses on a particular application without introducing any technical novelty, and the task is somewhat contrived for the application domain.

The task of ab initio reconstruction for homogeneous proteins is well-studied/solved. The authors should benchmark against a state of the art algorithm.

Fundamentally, the method relies on a training set of previously posed images, so the task they are addressing is not a realistic setting. As the authors discuss, if this approach were extended to be trained across multiple datasets and could predict orientations for a new (real) dataset zero-shot, it would be very high impact; however I am not convinced from the current results that extending this approach is promising. The paper lacks theoretical grounding, which the authors admit, and I am not convinced that there are any generalization properties for functions on distances between projection images (to unseen proteins).




**Time Spent Reviewing:**

3

---

> ### Author Response · Authors · 2021-08-09
> **Rebuttal to Review by Reviewer zuH5**
>
> Thank you for your time and thoughtful comments.
>
> We do think the paper is suitable for publication at NeurIPS. The novelty of our work lies in the tackling of a specific and important problem in computational biology with an innovative combination of (indeed existing) ML techniques; we believe this could open the door to new developments in ML for cryo-EM. Regarding the "contrivance for the application domain": The call for papers states that "NeurIPS 2021 is an interdisciplinary conference that brings together researchers in machine learning, [...], *computational biology*, and other fields" and specifically lists "Applications (e.g., speech processing, *computational biology*, computer vision, NLP)" as a topic of interest. Cryo-EM is one of the leading problems in computational biology nowadays, and its complexity obviously requires solutions that are tailored to the application domain. Hence, we do believe that our paper perfectly fits within the scope of NeurIPS.
>
> Regarding the request to further evaluate the method on unseen proteins and to compare it to existing pipelines: We refer the reviewer to our top-level "General Rebuttal" comment, where those questions are addressed.

---

### Author Response · Authors · 2021-08-09
**General Rebuttal**

As the reviewers have rightly pointed out (and as we discuss in Section 4), the applicability of the proposed method to real practical situations is still conditioned on demonstrating its accuracy on "unseen" proteins (transfer learning). Once this is achieved, an extensive comparison with the most commonly-established pipelines in the field (cryoSPARC, Relion, Aspire, etc.) is definitely required.

While it would have been ideal to deliver a new angle-refinement software for cryo-EM that is fully deployable in practice and competitive with the state-of-the-art, the task is a notoriously-challenging one: Cryo-EM measurements are some of the noisiest data in biomedical imaging, and the global algorithmic process equates to a high-dimensional non-convex optimization problem. As a consequence, most of the current well-known cryo-EM processing packages are themselves the result of years of iterative refinement (no pun intended).

At the present, we have focused on proposing a new paradigm for estimating the orientations in cryo-EM, and have provided a first demonstration of the feasibility of this method in an (admittedly) simplified cryo-EM setting. We see the extension of the applicability of the method to a wider range of data and the comparison to other existing packages as a natural follow-up of this work, which we hope will soon be addressed in a separate contribution; we provide some pointers for this in Section 4.

---

### Decision · Program_Chairs · 2021-09-27

**Decision:**

Reject

**Comment:**

The Cryo-EM problem is of significant importance and ML techniques for it are in scope for the NeurIPS conference.  As the submission did not provide a novel machine learning algorithm, the paper was evaluated primarily on its potential impact in application.  There was a consensus among the reviewers that the paper suffers from methodological concerns, including a lack of comparison to appropriate baselines given the same setup, the generalizability of the method toward proteins not seen in training data, and unclear performance improvement as part of a full cryo-EM pipeline.  Consequently, it is difficult to know how significant the result is in application.  The authors are encouraged to address these concerns in a future submission.